# Vibrational hierarchy leads to dual-phonon transport in low thermal conductivity crystals

Yixiu Luo [1,2], Xiaolong Yang[1,3], Tianli Feng[4], Jingyang Wang [2] & Xiulin Ruan [1✉]

Many low-thermal-conductivity ($\kappa_L$) crystals show intriguing temperature ($T$) dependence of $\kappa_L$: $\kappa_L \propto T^{-1}$ (crystal-like) at intermediate temperatures whereas weak $T$-dependence (glass-like) at high temperatures. It has been in debate whether thermal transport can still be described by phonons at the Ioffe-Regel limit. In this work, we propose that most phonons are still well defined for thermal transport, whereas they carry heat via dual channels: normal phonons described by the Boltzmann transport equation theory, and diffuson-like phonons described by the diffusion theory. Three physics-based criteria are incorporated into first-principles calculations to judge mode-by-mode between the two phonon channels. Case studies on $La_2Zr_2O_7$ and $Tl_3VSe_4$ show that normal phonons dominate low temperatures while diffuson-like phonons dominate high temperatures. Our present dual-phonon theory enlightens the physics of hierarchical phonon transport as approaching the Ioffe-Regel limit and provides a numerical method that should be practically applicable to many materials with vibrational hierarchy.

[1] School of Mechanical Engineering and the Birck Nanotechnology Center, Purdue University, West Lafayette, IN 47907, USA. [2] Shenyang National Laboratory for Materials Science, Institute of Metal Research, Chinese Academy of Sciences, Shenyang 110016, China. [3] Institute for Advanced Study, Shenzhen University, Shenzhen 518060, China. [4] Energy and Transportation Science Division, Oak Ridge National Laboratory, Oak Ridge, TN 37831, USA. ✉email: ruan@purdue.edu

ow-thermal conductivity ($\kappa_L$) crystals are of great interest in a variety of applications including thermal barrier coatings (TBC), thermoelectrics and nuclear reactors. They often show intriguing thermal-transport properties: $\kappa_L$ decreases inversely with temperature ($T$) at intermediate temperatures as expected for crystals; but shows weak or even no distinct dependence on $T$ at high temperatures, which is an anomalous, glass-like behavior[1–7]. While the former can be explained within the scheme of standard phonon Boltzmann transport equation (BTE) by primarily considering three-phonon scattering[8], the latter is still an open question. Advance in developing unified theories and numerical methods is just looming over the horizon to improve understanding the disparate $\kappa_L \sim T$ relationship in these crystals.

Recent studies have identified that in low-$\kappa_L$ crystals, some phonon modes have mean free path ($l$) shorter than the Ioffe–Regel limit[9], casting questions whether these modes can still be defined as phonons and how they contribute to thermal transport. Theoretical models of various sophistication have been developed to resolve this challenge[10–16]. Agne et al.[10] proposed that heat transfer in low-$\kappa_L$ complex crystals could be reasonably described by assuming a media of diffusons according to the random-walk-based diffusion theory, as opposed to phonons, and a model of diffuson-mediated $\kappa_L$ was proposed to better explain the experimental results particularly at high temperatures. The diffuson model by itself, however, does not fit to pure crystals where phonons are still well-defined as ensured by the periodicity of the lattice. On the other hand, other models have considered the hierarchy of vibrational modes based on the Ioffe–Regel limit. Chen et al.[11] proposed that for weakly disordered crystals with complex unit cell (e.g., higher manganese silicides), $\kappa_L$ could be explained by a hybrid phonon and diffuson model. The model employed a few fitting parameters and used the inelastic neutron scattering spectra to obtain an Ioffe–Regel crossover-frequency of 20 meV, below which the vibrational modes were treated as phonons and above which were diffusons. Mukhopadhyay et al.[12] used the scale of interatomic spacing as Ioffe–Regel criterion and proposed that for $Tl_3VSe_4$ crystals, the phonon modes with mean free path $l$ smaller than the Ioffe–Regel limit no longer behave as phonons, but should be replaced by hopping modes whose frequencies or eigenvectors cannot be meaningfully defined. They hence proposed a two-channel model that combines the phonon channel treated with BTE and the hopping channel calculated using Einstein's model or Cahill's model, yielding $\kappa_L$ and its temperature dependence in better agreement with experimental data. However, some open questions still remain. Particularly, how do the well-defined phonons interact with the hopping modes? How to subtract the well-defined modes from the hopping channel in $\kappa_L$ calculation? On the other hand, we note that infrared and Raman spectroscopies of some semiconductors have shown that, the zone-center optical phonons, many of which would have extremely short or nearly zero $l$, still have well-defined frequencies and linewidths (scattering rates) that can be accurately predicted by first-principles calculations[17,18]. This may suggest that the phonon concept for these modes is still valid, while the failure is with BTE which does not recognize the physical lower-limit of $l$. We can also note that this is not the first time BTE fails for phonons; in fact, BTE was known to fail for coherent phonons in superlattices or phononic crystals; therein, it does not capture the wave effects[19,20].

Most recently, attempt has also been made to unify the thermal-transport theory in crystals and glasses. Simoncelli et al.[13] have transmuted the BTE formulism into a $\kappa_L$ equation written in terms of the phonon velocity operator with diagonal and off-diagonal elements describing the particle-like propagation of phonons and the wave-like tunneling of coherence,

respectively. Applying this model in perovskite $CsPbBr_3$ arrives at reasonable simulation of its glass-like $\kappa_L$. Meanwhile, Isaeva et al.[14] developed a quasi-harmonic Green–Kubo method, as if to generalize the Allen–Feldman model[21–24] for amorphous systems into relaxation-time-approximation-based BTE for crystals by expressing energy transport in a quantum mechanical fashion, and gives reasonable $\kappa_L$ predictions for amorphous and crystalline Si. Nevertheless, further physical insights are expected to improve understanding the nature of hierarchical vibrations in the context of physically based theories.

Inspired by the idea of vibrational hierarchy from previous models, while attempting to resolve the open questions, in this work we sparkle a different concept that the vibrational modes with very short $l$ could still be treated as phonons with well-defined frequencies, eigenvectors, and scattering rates, but their heat conduction should be described by the diffusion theory instead of BTE. We hence propose a dual-phonon theory, by treating the short-$l$ phonons with the diffusion theory, and other normal phonons with BTE. Our theory does not introduce a different type of heat-carrying modes other than phonons, and eliminates the theoretical challenge of how they would interact with phonons and alter their scattering events. Also, for a sophisticated model, we introduce three different criteria, based on phonon mean free path, wavelength, and thermal diffusivity, to judge mode-by-mode between the normal phonon channel and diffuson-like phonon channel, and hence avoid the double-counting issue. The three criteria all yield consistent results that agree quantitatively with experiments, demonstrating the robustness and predictive capability of our theory. Our theory is demonstrated on $La_2Zr_2O_7$, a thermal-barrier-coating (TBC) candidate material for gas turbine technologies[25], and $Tl_3VSe_4$, a potential thermoelectric material[12]. More background of $La_2Zr_2O_7$ could be found in Supplementary Note 1. Our approach is able to explain the thermal conductivity and the intriguing $\kappa_L \sim T$ dependence over the entire temperature range, and is expected to help understand the thermal transport of such low-$\kappa_L$ crystals. Also, our model may provide physical insights toward unifying the theories of thermal conductivity in crystals and amorphous materials.

## Results

**Normal phonons and diffuson-like phonons.** We first calculate the phonon properties of $La_2Zr_2O_7$ using the standard anharmonic lattice dynamics based on density functional theory. Immediately we find that a large percentage of vibrational modes have very small $l$ even at room temperature, and more so at high temperatures, as shown in Fig. 1a. Similar features have been identified for $Tl_3VSe_4$ (see Supplementary Fig. 5). Apparently, these modes cannot be treated as normal phonons, which by definition, are expected to propagate far enough to sample the periodicity of the transport media, i.e., comparable to the scale of phonon wavelength or several lattice spacing[21–23]. In developing our theory, these modes are treated as diffuson-like phonons.

In order to sophisticatedly judge whether a phonon mode should be treated as normal phonon or diffuson-like phonon, we propose three criteria, as conceptually shown in Fig. 2. The first two criteria are derived from the Ioffe–Regel limit of phonons[9], arguing that the value of $l$ for normal phonons should not be smaller than their wavelength ($\lambda$) (I. $l$–$\lambda$ criterion), or the minimum interatomic spacing ($a_{min}$) of the lattice (II. $l$–$a_{min}$ criterion), respectively. The third criterion argues that a vibrational mode should be characterized as diffuson-like if its phonon thermal diffusivity ($D_{Phon}$) becomes smaller than its diffuson thermal diffusivity ($D_{Diff}$) (III. $D_{Phon}$–$D_{Diff}$ criterion). Here we note that, another reasonable criterion to define a

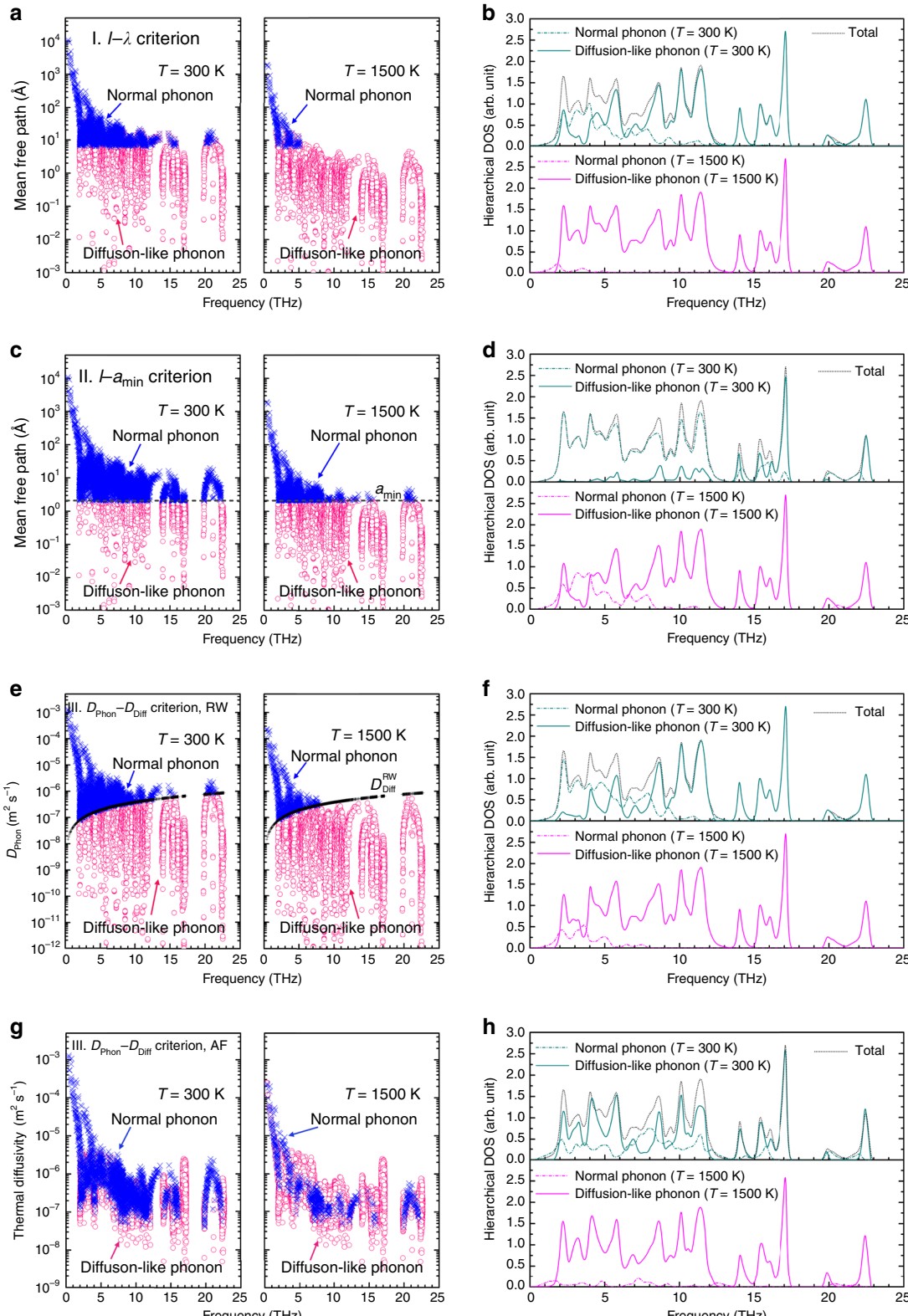

**Fig. 1 Hierarchy of lattice vibrations in La₂Zr₂O₇.** The vibrational modes are assigned as normal phonons or diffuson-like phonons according to **a** Criterion I, whether their mean free path ($l$) is above or below the vibrational wavelength ($\lambda$); **c** Criterion II, whether their $l$ is above or below the minimum interatomic spacing ($a_{min} = 2.1$ Å for La₂Zr₂O₇, drawn in dashed line); **e** Criterion III, whether the phonon thermal diffusivity ($D_{Phon}$) is above or below the diffuson thermal diffusivity ($D_{Diff}$), in which $D_{Diff}$ is calculated based on the random-walk theory ($D_{Diff}^{RW}$) and plotted as a function of the vibrational frequency in black marks; and **g** Criterion III, with $D_{Diff}$ calculated based on the Allen–Feldman formula ($D_{Diff}^{AF}$); herein, the value of $\max\left[D_{Phon}, D_{Diff}^{AF}\right]$ for each vibrational mode is plotted. The calculated hierarchical phonon density of states (DOS) for normal phonons and diffuson-like phonons according to **b** Criterion I; **d** Criterion II; **f** Criterion III coupled with the random-walk (RW) theory; and **h** Criterion III coupled with the Allen–Feldman (AF) theory. Note: the ordinary frequency ($\nu = \omega/2\pi$) in units of THz is used for illustration. Source data are provided as a Source data file.

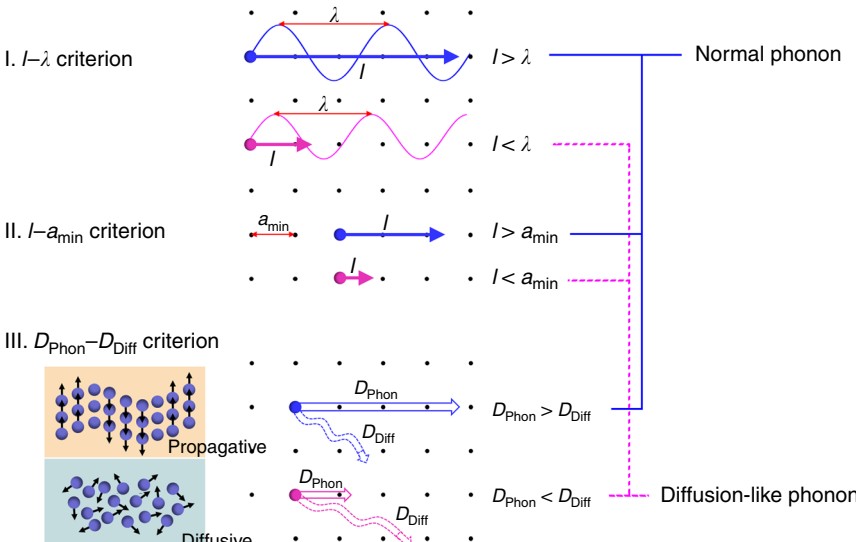

**Fig. 2 Dual-phonon judging criteria for vibrational modes.** Three different criteria are proposed for the assignment of each vibrational mode to be a normal phonon or a diffuson-like phonon. The visual representation of the diffusive/propagative transport is drawn in analogy with Agne et al.'s work[10].

normal phonon mode is based on its long-enough relaxation time ($\tau$) as $\tau \gg 1/\nu$ ($\nu$ is the vibrational frequency). For acoustic phonons approaching the long-wavelength-limit, this criterion is the same with the $l$–$\lambda$ criterion, whereas for some phonons whose phase velocity ($\nu_p$) is considerably higher than group velocity ($\nu_g$), it is less restrictive[23]. Therefore, the $\tau$-criterion is not considered in developing our theory.

For modeling $\kappa_L$ in materials with such strong vibrational hierarchy, we assume a thermal-transport media with both normal phonons and diffuson-like phonons. All these phonon modes are able to transfer heat (i.e., non-localized), but following BTE theory for the former and diffusion theory for the latter, respectively. Our dual-phonon theory combines a normal phonon channel and a diffuson-like phonon channel:

$$\kappa_L^{Dual-phonon} = \kappa_L^{Phon} + \kappa_L^{Diff} \qquad (1)$$

Heat conduction from both channels are derived by following the physical picture that heat is carried by atomic vibrations of a solid. There are three components to be considered: (1) the number of vibrational modes that are available to carry heat, summed up to be the total degrees of vibrational freedom; (2) the amount of heat that can be carried by each vibrational mode; (3) the propagation behavior of vibrations through the dual-phonon media. We write the contribution from each channel as:

$$\kappa_L^{Phon} = \sum_{i=1}^{N_{Phon}} C_s(i) D_{Phon}(i) \qquad (2)$$

$$\kappa_L^{Diff} = \sum_{j=1}^{N_{Diff}} C_s(j) D_{Diff}(j) \qquad (3)$$

Here, $i$ and $j$ denote the indices of vibrational modes, sampled over the Brillouin zone (BZ). $N_{Phon}$ and $N_{Diff}$ are the numbers of normal phonons and diffuson-like phonons, respectively. One of the merits of our theory is that, we impose the conservation of vibrational degrees of freedom by requiring $N_{Phon} + N_{Diff} = 3N$, where $N$ is the total number of atoms. As such, once a mode is characterized as a normal phonon, it cannot be treated as a diffuson-like phonon at the same time, and vice versa. $C_s$ is the per-mode specific heat following the Bose–Einstein statistics of phonons. $D_{Phon}$ and $D_{Diff}$ are defined as the per-mode thermal diffusivities for normal phonons and diffuson-like phonons,

respectively. In our present model, $D_{Phon}$ is calculated using the phonon BTE theory; whereas $D_{Diff}$ could be calculated based on the random-walk theory[10] (denoted as $D_{Diff}^{RW}$) or the Allen–Feldman theory[23] (denoted as $D_{Diff}^{AF}$). Details are presented in the Methods section. In the present work, we only performed limited test for our dual-phonon theory coupled with $D_{Diff}^{AF}$ (using the III. $D_{Phon}$–$D_{Diff}$ criterion), mainly due to its high computational expense.

**Hierarchy of lattice vibration in La$_2$Zr$_2$O$_7$.** The hierarchy of lattice vibrations for La$_2$Zr$_2$O$_7$ is shown in Fig. 1. We see that the vibrational modes that are judged as diffuson-like phonons account for 75.35%, 23.82%, 65.42%, and 69.73% of the total number of modes at $T = 300$ K, respectively, according to our proposed judging criteria I, II, and III coupled with the random-walk theory, and III coupled with the Allen–Feldman theory. The fractions rise to 98.00%, 80.23%, 91.25%, and 93.55% at $T = 1500$ K. As illustrated in Fig. 3, this behavior originates either from small $\nu_g$, especially for high-frequency vibrations, or from small $\tau$, indicative of intense scattering among those modes. The increased population of diffuson-like phonons at higher temperatures is presumably due to increased phonon scattering.

To better visualize the degree of the vibrational hierarchy, the vibrational density of states (DOS) for normal phonons vs. diffuson-like phonons are illustrated in Fig. 1b, d, f, h. There is a crossover from normal phonon-dominated states at the low-frequency range to diffuson-like phonon-dominated states at high-$\nu$ range, and the crossover shifts to lower-$\nu$ at higher temperatures. Take the case of II. $l$–$a_{min}$ Criterion as an example. At $T = 300$ K, normal phonons dominate below $\nu \sim 13$ THz; whereas for $T = 1500$ K, diffuson-like modes exhibit the first peak at $\nu \sim 2$ THz, and quickly becomes dominant at above $\nu \sim 5$ THz. This could be understood from increased scattering at higher temperatures, resulting in suppressed $\tau$ and $l$ values for all vibrational modes. Furthermore, detailed comparisons among the three criteria show that, the number of vibrational modes assigned as diffuson-like phonons based on the random-walk theory increases in the order of II. $l$–$a_{min}$ Criterion, III. $D_{Phon}$–$D_{Diff}$ Criterion, and I. $l$–$\lambda$ Criterion, and the major difference comes from assignment of the vibrational modes in the frequency range $2 < \nu < 13$ THz. On the one hand, the fact that a larger number of vibrations are assigned as diffuson-like phonons by Criterion III than Criterion II could be understood from the interplay of low

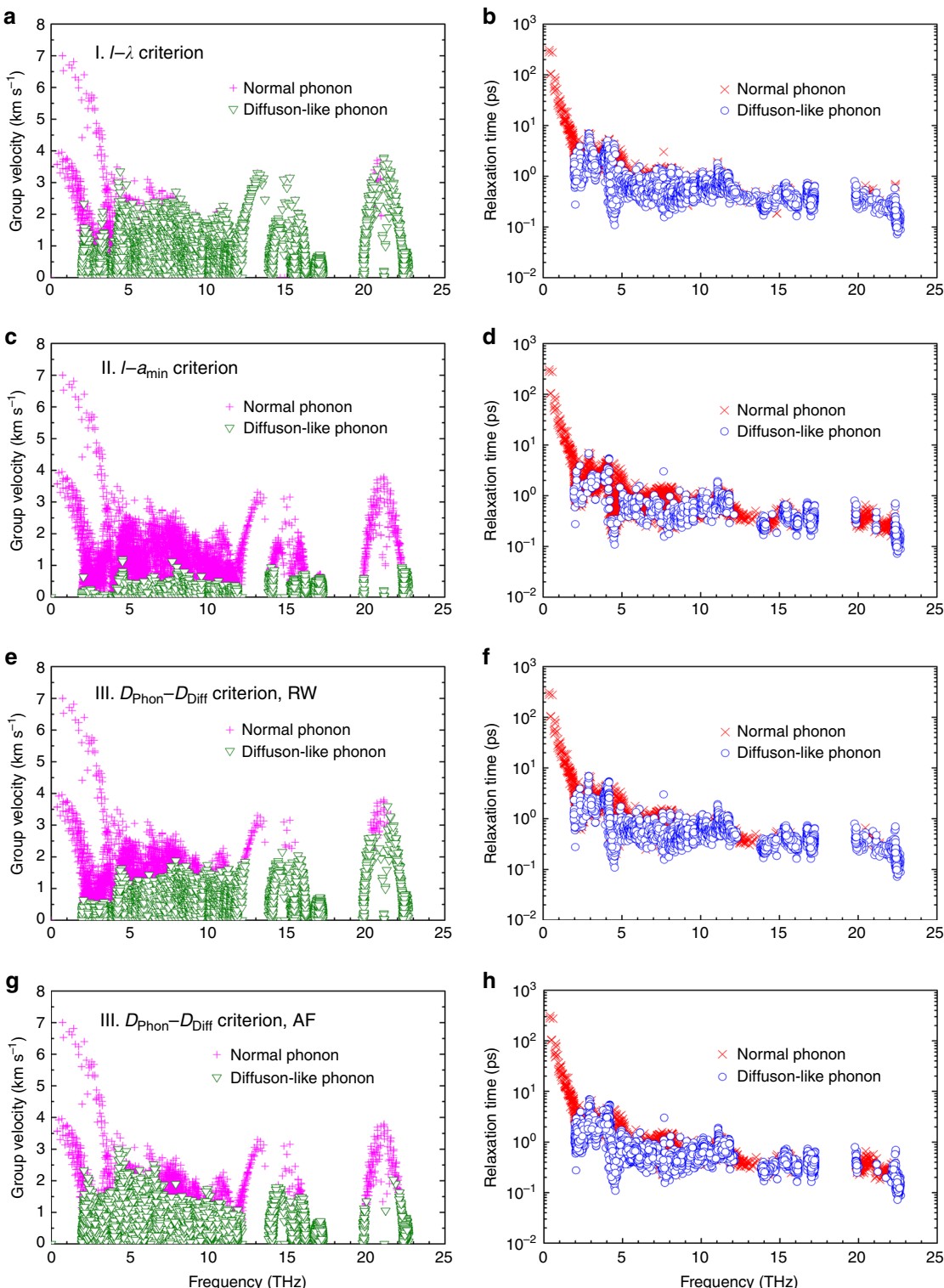

**Fig. 3 Hierarchy of phonon group velocity and relaxation time.** The calculated (**a**), (**c**), (**e**), (**g**) phonon group velocity ($v_g$) and (**b**), (**d**), (**f**), (**h**) phonon relaxation time ($\tau$) at $T = 300$ K for each vibrational mode of $La_2Zr_2O_7$, based on criteria I, II, and III coupled with the random-walk (RW) theory, and III coupled with Allen–Feldman (AF) theory, respectively. Source data are provided as a Source data file.

$v_g \cdot l$ and high $v$ for certain modes. On the other hand, the difference between Criterion I and Criterion II is expected to be more pronounced for zone-center long-wavelength vibrations, as certain modes have $a_{min} < l < \lambda$ hence they are assigned as normal phonons by the $l$–$a_{min}$ Criterion while as diffuson-like phonons by the $l$–$\lambda$ Criterion. Moreover, as shown in Fig. 1f,

the ratio $N_{Diff}/N_{Phon}$ appears to increase with the vibrational frequency under Criterion III coupled with the random-walk theory, but the trend is not so clear for the results derived from the Allen–Feldman theory (Fig. 1h). Such difference could be understood from the different frequency dependence of $D_{Diff}^{RW}$ vs. $D_{Diff}^{AF}$. The random-walk theory assumes that heat is

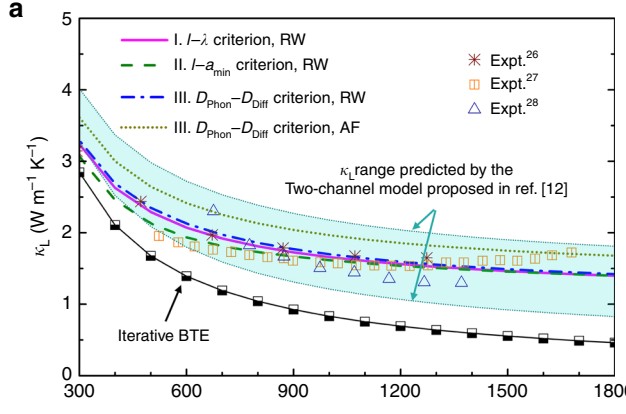

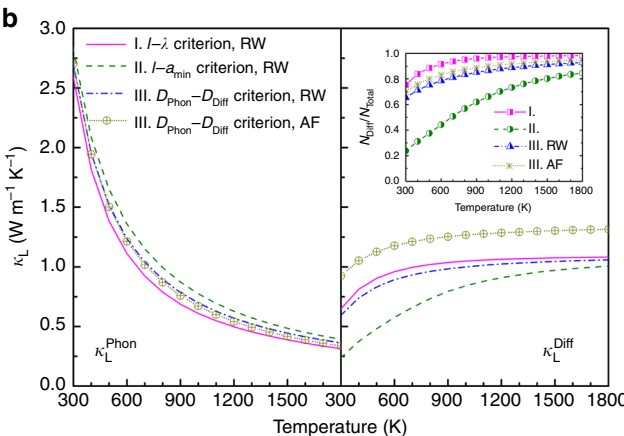

**Fig. 4 Lattice thermal conductivity of La₂Zr₂O₇. a** The calculated temperature-dependent lattice thermal conductivity ($\kappa_L$) for La₂Zr₂O₇ using our dual-phonon theory. The results from the iterative solution of BTE, Mukhopadhyay et al.'s two-channel model[12], and experiments[26–28] are presented for comparison. **b** The contribution to total $\kappa_L$ from normal phonons ($\kappa_L^{Phon}$) and diffuson-like phonons ($\kappa_L^{Diff}$). Inset: The number of diffuson-like phonons ($N_{Diff}$) divided by the number of all vibrational modes ($N_{Total} = N_{Phon} + N_{Diff}$) for La₂Zr₂O₇, calculated as a function of temperature. Source data are provided as a Source data file.

transferred through successful random jumps of independent oscillators within a period of oscillation to a distance related with the number density of atoms, and thus arrives at $D_{Diff}^{RW} \propto \nu$ throughout the spectrum. The Allen–Feldman theory, on the other hand, is rooted in the thermal correlation of each vibrational mode and arrives at varied frequency dependence of $D_{Diff}^{AF}$. See Supplementary Note 2 and Supplementary Fig. 1 for more details.

**Lattice thermal conductivity.** The $\kappa_L$ values calculated from an iterative solution of BTE coupled with anharmonic lattice dynamics are shown in Fig. 4a. Both phonon–phonon scattering and isotope scattering are considered in our calculations, but the latter has little impact on $\kappa_L$ values and their temperature dependence. At room temperature, the calculated $\kappa_L$ shows reasonable agreement with experimental data[26–28]. At higher temperatures, it yields $\kappa_L \sim T^{-1}$ dependence as expected from the dominance of phonon–phonon Umklapp scattering, whereas experimental data goes nearly temperature-independent. The calculated $\kappa_L = 0.55 \text{ W·m}^{-1}\text{·K}^{-1}$ at $T = 1500$ K is only one-third of the measured value $\kappa_L = 1.5 \text{ W·m}^{-1}\text{·K}^{-1}$, and is even lower than the reported high-temperature limit ($\kappa_{min} = 1.2 \text{ W·m}^{-1}\text{·K}^{-1}$) for La₂Zr₂O₇ crystals[29,30]. Moreover, we notice that including the

state-of-the-art four-phonon scattering[31–33] is expected to give further reduction of $\kappa_L$ at high temperature and results in stronger-than-$T^{-1}$ temperature dependence, and thus could not bridge the gap between experimental and BTE-derived results in our case. Guided by the Ioffe–Regel limit of phonons[9], we recalculate $\kappa_L$ within the framework of phonon BTE theory, by assuming a lower-bound of $l$ for all modes, i.e., the $l$ values are manually set to $a_{min}$ if $l < a_{min}$. This trial yields $\kappa_L = 0.61 \text{ W·m}^{-1}\text{·K}^{-1}$ at $T = 1500$ K, showing inadequate correction to the previous results. Such significant gap between the calculated temperature-dependent $\kappa_L$ and experimental values indicate that the conventional BTE theory becomes invalid for heat conduction in La₂Zr₂O₇, and a different physical picture is needed to describe the transport behavior of the interesting small-$l$ or small-$D_{Phon}$ phonon modes.

Next, we calculate $\kappa_L$ of La₂Zr₂O₇ using the two-channel model proposed by Mukhopadhyay et al.[12], by combining a phonon conduction channel ($\kappa_{phonon}$) and a hopping channel ($\kappa_{hop}$) from modes of $l < a_{min}$. In this model, $\kappa_{phonon}$ is calculated from phonon BTE theory, by excluding the contribution from small-$l$ vibrations; whereas $\kappa_{hop}$ is calculated either via Cahill's formula[34,35] or Einstein's formula from which the Cahill's model is derived. Cahill's formula requires $v_g$ of acoustic phonons as input parameters, which could be extracted from phonon dispersions of La₂Zr₂O₇ (shown in Supplementary Fig. 2): 3837 and 3969 m·s⁻¹ for transverse acoustic phonons (TA1 and TA2), and 6872 m·s⁻¹ for longitudinal acoustic phonons (LA). Einstein's formula requires defining the Einstein temperature ($\theta_E$) for the oscillators, which could be estimated from the calculated low-temperature specific heat of La₂Zr₂O₇ (shown in Supplementary Fig. 3): $\theta_E = 180$ K, in moderate agreement with reported values[36]. Using these parameters as input, the total $\kappa_L$ ($\kappa_{phonon} + \kappa_{hop}$) are calculated to be 1.90 and 0.92 W·m⁻¹·K⁻¹ at $T = 1500$ K by employing Cahill's formula and Einstein's formula, respectively. The corresponding temperature dependencies are $\kappa_L \sim T^{-0.31}$ and $\kappa_L \sim T^{-0.65}$ above $T = 1000$ K. Now the experimental values of $\kappa_L$ fall within the range provided by the two-channel model, and the temperature dependence shows improved agreement. As shown in Fig. 4a, the range in between might be originated from uncertainties in defining the $\theta_E$ values, which has also been pointed out in ref. [12]. Besides, we note that in order to preserve the total number of vibrational modes, the normal phonon modes need to be subtracted from the hopping channel, which is difficult to do when incorporating the Cahill's formula in the two-channel model.

The results of our dual-phonon theory for La₂Zr₂O₇ are shown in Fig. 4a. Our model coupled with the random-walk theory gives weakened $\kappa_L \sim T$ dependence as $T$ increases: $\kappa_L \sim T$ relationship calculated using I. $l - \lambda$ criterion (II. $l - a_{min}$ criterion; III. $D_{Phon} - D_{Diff}$ criterion) weakens from $\kappa_L \sim T^{-0.69}$ ($\kappa_L \sim T^{-0.72}$; $\kappa_L \sim T^{-0.67}$) for 300 K < $T$ < 500 K, to $\kappa_L \sim T^{-0.47}$ ($\kappa_L \sim T^{-0.40}$; $\kappa_L \sim T^{-0.47}$) for 500 K < $T$ < 1000 K, and to $\kappa_L \sim T^{-0.29}$ ($\kappa_L \sim T^{-0.24}$; $\kappa_L \sim T^{-0.30}$) for $T > 1000$ K, reasonably reproducing the flattening-out behaviors of the experimental $\kappa_L \sim T$ data. Partial contributions from normal phonons ($\kappa_L^{Phon}$) and diffuson-like phonons ($\kappa_L^{Diff}$) to the total $\kappa_L$ are plotted in Fig. 4b. It is shown that, $\kappa_L^{Diff}$ starts to dominate over $\kappa_L^{Phon}$ at around $T = 700$, 1000, and 800 K based on Criterion I, II, and III, respectively. Clearly, the role of diffuson-like phonons for the heat conduction of La₂Zr₂O₇ is more significant at high-temperature ranges. Besides, the calculated $\kappa_L^{Phon}$ decreases in the order of criteria II, III, and then I, while the trend is reversed for $\kappa_L^{Diff}$. This result is consistent with the percentage of diffuson-like phonons assigned out of all the vibrational modes (inset Fig. 4b). Evidently, despite using three different criteria to judge normal phonons vs. diffuson-like phonons, the calculated $\kappa_L$ values as

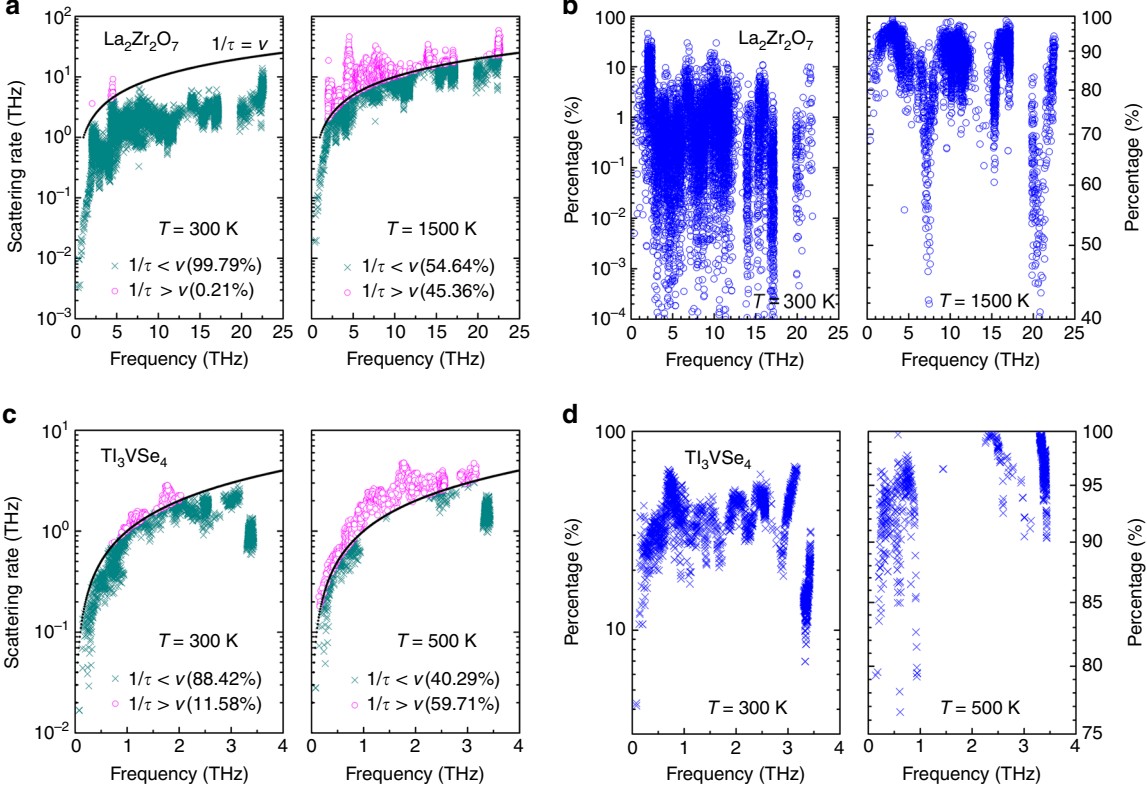

**Fig. 5 Contributions from the ill-defined-in-time phonons. a, c** The scattering rates ($1/\tau$) of $La_2Zr_2O_7$ and $Tl_3VSe_4$ calculated from standard anharmonic lattice dynamics. The results at $T = 300\,K$ and $T = 1500\,K$ are presented for $La_2Zr_2O_7$, and $T = 300\,K$ and $T = 500\,K$ for $Tl_3VSe_4$. The vibrational modes having $1/\tau < \nu$ (well-defined in the time scale) and $1/\tau > \nu$ (ill-defined in the time scale) are distinguished in different colors; and the baseline $1/\tau = \nu$ is plotted as a guide for the eye. **b, d** The percentages of scattering rates of well-defined-in-time modes contributed by three-phonon scattering processes involving at least one ill-defined-in-time mode. Source data are provided as a Source data file.

well as the $\kappa_L \sim T$ dependence show quite similar features, demonstrating the robustness of our approach. Moreover, our dual-phonon theory is also validated on $Tl_3VSe_4$, a potential thermoelectric material with ultra-low $\kappa_L$ that was used as the model materials in ref. [12]. See Supplementary Note 3, Supplementary Figs. 4–7, and Supplementary Table 1 for detailed results and discussions.

Also shown in Fig. 4 are the results from our dual-phonon theory coupled with the Allen–Feldman model. The percentage of diffuson-like phonons assigned out of all the vibrational modes using the III. $D_{Phon}-D_{Diff}$ criterion coupled with $D_{Diff}^{AF}$ is generally consistent with the results derived with $D_{Diff}^{RW}$, and the calculated values of $\kappa_L^{Phon}$ from the two routes show minor differences. Moreover, the $\kappa_L \sim T$ dependence calculated with $D_{Diff}^{AF}$ gives $\kappa_L \sim T^{-0.61}$ for $300\,K < T < 500\,K$, $\kappa_L \sim T^{-0.42}$ for $500\,K < T < 1000\,K$, and $\kappa_L \sim T^{-0.27}$ for $T > 1000\,K$, consistent with that derived using $D_{Diff}^{RW}$. These results demonstrate the robustness of the dual-phonon theory. Evidently, imposing the dual-phonon theory coupled with Allen–Feldman model tends to yield higher $\kappa_L^{Diff}$, and thus slightly higher values of total $\kappa_L$, as compared with those derived with the random-walk picture. This is actually inherited from the different frequency-dependent formulas of $D_{Diff}^{RW}$ vs. $D_{Diff}^{AF}$ as we have discussed above. Interestingly, at least for $La_2Zr_2O_7$, the random-walk-based thermal diffusivities in our dual-phonon theory perform better than the Allen–Feldman theory. Meanwhile, this route requires far less computational expense, which is potentially favorable for high-throughput $\kappa_L$ predictions. Nevertheless, more works are required in the future to test this issue using a larger material pool.

## Discussion
We can see that our dual-phonon theory emphasizes the physics of hierarchical phonon transport in low-$\kappa_L$ materials, by treating all vibrational modes within the phonon picture yet with different thermal-transport behaviors, i.e., normal phonons with the BTE theory and diffuson-like phonons with the diffusion theory. Basically, our dual-phonon theory proposes a conceptual change that the vibrational modes are still phonons upon approaching the Ioffe–Regel limit, and the phonon frequency, eigenvector, and relaxation time could be rigorously described by first principles. Specifically, the identified diffuson-like phonons with small $l$ or small $D_{Phon}$ still fall in the physical picture of phonons with well-defined scattering rates that can be reliably predicted from first principles; just that their heat conduction cannot be treated using the scheme of mean free path or BTE. In this way, the scattering and thermal transport of diffuson-like phonons are decoupled. The three proposed physics-based judging criteria, without relying on any fitting parameters, enable a per-mode-based judging between normal phonon and diffuson-like phonon channels, and give consistent $\kappa_L$ predictions in agreement with experimental data, demonstrating the robustness of our approach. Moreover, while this paper was under review, we learned that progressive efforts have been made to unify thermal transport in crystals and amorphous materials, which involve sophisticated analytical and mathematical derivations, yet the corresponding physical insights are still under development[13,14]. In parallel with these efforts, our dual-phonon theory provides physical insights in understanding the vibrational hierarchy of crystals having low and glass-like $\kappa_L$, and leads up to a $\kappa_L$-prediction method with good robustness and favorable computational expense.

It would be interesting to explore future opportunities to expand and advance our proposed dual-phonon theory. First, the current study does not include the effects of four-phonon scattering or phonon renormalization, which are important effects addressed elsewhere but are beyond the scope of the present study, partly due to the very high computational cost for La$_2$Zr$_2$O$_7$ (with complex and large unit cell). Four-phonon scattering is important at high temperature for all materials and even at room temperature for strongly anharmonic materials and materials with exceptionally weak three-phonon scattering[31–33]. Phonon renormalization is important for strongly anharmonic materials, especially those exhibiting dynamic instability[37–40]. Indeed, consideration of four-phonon effect and the temperature-effect might arrive at more accurate simulation of the phonon group velocities and scattering, leading to better placement of normal phonons vs. diffuson-like phonons, and more accurate $\kappa_L$ and its temperature dependence.

Second, the primary scope of our dual-phonon theory is to differentiate the thermal-transport behaviors of normal phonons against diffuson-like phonons in crystals based on whether or not they are ill-defined in the space scale (in terms of small $l$ or $D_{Phon}$), given that they fall within the phonon picture in terms of scattering. Interestingly, we find that some vibrational modes in La$_2$Zr$_2$O$_7$ and Tl$_3$VSe$_4$ exhibit $1/\tau > \nu$; in other words, their lifetime is too short and thus should be considered ill-defined in the time scale. Similar behavior has been universally observed in crystals having certain types of perturbation or disorder[12,13,41–44]. As shown in Fig. 5a, c, the number of modes that are ill-defined in the time scale ($1/\tau > \nu$) account for 0.21% for La$_2$Zr$_2$O$_7$ and 11.58% for Tl$_3$VSe$_4$ at $T = 300$ K, which are much smaller than the percentages of the ill-defined-in-space phonons. They rise to 45.36% at $T = 1500$ K for La$_2$Zr$_2$O$_7$ and 59.71% at $T = 500$ K for Tl$_3$VSe$_4$, respectively, though. We then further evaluate the effects of these modes on the scattering rates of phonons that are well-defined in the time scale ($1/\tau < \nu$), by calculating the percentages of scattering rates of well-defined-in-time modes contributed by three-phonon scattering processes involving at least one ill-defined-in-time mode. Figure 5b, d shows that the percentages are mostly below 20% for La$_2$Zr$_2$O$_7$ and below 60% for Tl$_3$VSe$_4$ at $T = 300$ K, which rise to mostly above 70% at $T = 1500$ K for La$_2$Zr$_2$O$_7$, and above 90% at $T = 500$ K for Tl$_3$VSe$_4$. Therefore, although our approach makes a step closer to a sound theory and appears to agree with experiments, whether we can treat scattering by these ill-defined-in-time phonons with standard anharmonic lattice dynamics still remains a challenging open question, especially at high temperatures. Future studies are warranted on how to better understand the interactions between the well-defined-in-time phonons and ill-defined-in-time phonons in our model and other lately developed models[13,14].

The present results have important implications for a wide variety of low-$\kappa_L$ materials, including thermal barrier coatings (TBC), thermoelectrics, and nuclear materials. For these low-$\kappa_L$ materials, there might be large number of diffuson-like phonons characterized by small $l$ or small $D_{Phon}$, and the significant contribution from $\kappa_L^{Diff}$ to total $\kappa_L$ manifest with increased temperature, and eventually dominate over the contribution from $\kappa_L^{Phon}$ when approaching the high-temperature limit. In the context of our analysis on the TBC material La$_2$Zr$_2$O$_7$, the diffuson-like phonons mainly stem from low $\nu_g$ and/or high scattering rates, which could be further linked with the complexity of crystal structure and the heterogeneity of interatomic bonds. Such structural characteristics result in folding-in of phonon dispersion, suppression of phonon frequencies and thus group velocities, and serious tangling and scatterings among low-frequency acoustic and optical phonons (see Supplementary Note 4 and Supplementary Fig. 8 for details). This mechanism is expected to be a signature for many TBC candidates, i.e., oxide ceramics with complex crystal structure and vibrational hierarchy, probably including other RE$_2$Zr$_2$O$_7$ pyrochlores[25], silicates[45,46], and acuminate-silicates[47] etc. In this sense, our proposed dual-phonon model is expected to inspire deeper understanding and practical calculation methodology of $\kappa_L$ for TBC materials (see Supplementary Note 5 for details).

In summary, a dual-phonon theory is proposed for the $\kappa_L$ of crystals with vibrational hierarchy, by considering normal phonons described by the BTE theory and diffuson-like phonons described by the diffusion theory. Three physics-based criteria are used to judge mode-by-mode between normal phonons and diffuson-like phonons. Applying this theory on La$_2$Zr$_2$O$_7$ and Tl$_3$VSe$_4$ shows that $\kappa_L$ is mainly contributed by normal phonons at low temperatures, whereas by diffuson-like phonons at high temperatures. This theory successfully predicts the flattening-out of $\kappa_L \sim T$ trend upon temperature increment, in much better agreement with experiments than the conventional BTE theory. Meanwhile, it resolves the limitations of other existing models and leads to a computational procedure showing promising applicability. The improvement of heat conduction theory in low-$\kappa_L$ crystals will provide important insights in the development of TBC materials, thermoelectric materials, and nuclear materials. Future studies can include a comparison of our approach with those in refs. [13,14], a consideration of how to combine the mathematical rigor of those approaches and the physical insights from our approach and other heuristic models, and a better understanding of the interactions between the well-defined-in-time phonons and ill-defined-in-time phonons.

## Methods

**Calculation details of the dual-phonon theory.** In this study, the per-mode specific heat is calculated by following the Bose–Einstein statistics for phonon:

$$C_s(i;j) = \frac{1}{V} k_B \left( \frac{\hbar \omega(i;j)}{k_B T} \right)^2 \frac{\exp(\hbar \omega(i;j)/k_B T)}{[\exp(\hbar \omega(i;j)/k_B T) - 1]^2} \quad (4)$$

where $k_B$ is the Boltzmann constant; $\hbar$ is the reduced Planck constant; $T$ is the absolute temperature; $\omega$ is the angular frequency of a vibrational mode; $V$ is the volume of simulated unit cell.

$D_{Phon}$ is calculated using the phonon BTE theory, by incorporating the phonon group velocity ($\nu_g$), mean free path ($l$), and relaxation time ($\tau$) resulted from the scattering process. $D_{Diff}$ could be calculated by following the random-walk scheme proposed in Agne et al.'s model[10], which assumes that heat is transferred by discrete jumps between independent harmonic oscillators. Its linear dependence with respect to $\omega$ could be understood from the diffusive picture, as if oscillators jumping at more steps per time-interval contribute to higher rates of energy transfer. Basically, the diffusion term $D_{Diff}$ has the units of $D_{Phon}$ (m$^2\cdot$s$^{-1}$), while avoiding independent definition of $\nu_g$ and $l$ in the diffuson picture.

$$D_{Phon}(i) = \frac{1}{3} \nu_g(i) l(i) \quad (5)$$

$$l(i) = \nu_g(i) \tau(i) \quad (6)$$

$$D_{Diff}^{RW}(j) = \frac{1}{3} \frac{n^{-2/3} \omega(j)}{\pi} \quad (7)$$

where $n$ is the number density of atoms of the unit cell. In this study, mode-resolved vibrational parameters ($\nu_g$, $l$, $\lambda$, $\omega$, $\tau$, etc.) are gathered from outputs of DFT-based harmonic and anharmonic lattice dynamics calculations. Scattering information ($\tau$ and $l$) is gathered from iterative solution of the BTE; however, we tested that using parameters derived under the relaxation-time approximation (RTA) would have negligible influence on the present results.

For comparison, $D_{Diff}$ could also be calculated using the Allen–Feldman formula[23], which is grounded in the Green–Kubo theory and is at the reach of ab initio simulations.

$$D_{Diff}^{AF}(j) = \frac{\pi V^2}{3\hbar^2 \omega(j)^2} \sum_{p \neq j} \left| \mathbf{S}_{jp} \right|^2 \delta(\omega(j) - \omega(p)) \quad (8)$$

where $\mathbf{S}_{jp}$ is the heat-current operator measuring the thermal coupling between vibrational mode $j$ and $p$ based on their frequencies and spatial overlap of eigenvectors, and could be calculated from harmonic lattice dynamics. $\delta$ is the Dirac Delta function that could be approximated using Lorentzian broadening of

width greater than the average mode frequency interval ($\Delta_{\text{avg}}$).

$$\mathbf{S}_{jp} = \frac{\hbar}{2V} \mathbf{v}_{\mathbf{K}jp} (\omega_{\mathbf{K}}(j) + \omega_{\mathbf{K}}(p)) \tag{9}$$

$$\mathbf{v}_{\mathbf{K}jp} = \frac{1 \cdot i}{2\sqrt{\omega_{\mathbf{K}}(j)\omega_{\mathbf{K}}(p)}} \sum_{\alpha\beta} \sum_{s,k,k'} e_\alpha(k;\mathbf{K},j) D_{\beta\alpha}^{k'k}(0,s) \\ \times (\mathbf{R}_s + \mathbf{R}_{kk'}) e^{i\mathbf{K}\cdot(\mathbf{R}_s + \mathbf{R}_{kk'})} \times e_\beta(k';\mathbf{K},p) \tag{10}$$

where the wave vector $\mathbf{K}$ is summed over the Brillouin zone; $e$ is the corresponding eigenvector; $\alpha$ and $\beta$ denote the Cartesian directions. $\mathbf{R}_s$ is the distance between each unit cell (labeled $s$) and the basis unit cell (labeled 0) within a periodic supercell system; and $\mathbf{R}_{kk'}$ is the distance between atom $k$ and atom $k'$ within a unit cell. $D_{\beta\alpha}^{k'k}(0,s)$ is the dynamical matrix element, derived from the second-order interatomic force constant ($\Phi_{\beta\alpha}^{k'k}(0,s)$).

$$D_{\beta\alpha}^{k'k}(0,s) = \Phi_{\beta\alpha}^{k'k}(0,s)/\sqrt{m_k m_{k'}} \tag{11}$$

where $m_k$ and $m_{k'}$ are the masses of the atom $k$ and $k'$.

**Computation details for first principles.** Density functional theory (DFT) calculations on $La_2Zr_2O_7$ are performed using the projected augmented wave (PAW) method[48] as implemented in Vienna Ab Initio Simulation Package (VASP)[49] with electronic exchange and correlations treated in the localized density approximation (LDA)[50]. The wave functions are expanded in a plane-wave basis with the kinetic energy cutoff of 600 eV, and Monkhorst–Pack[51] Γ-centered $k$-mesh of $7 \times 7 \times 7$ is used to sample the Brillouin zone (BZ). Cell parameters and internal atomic positions are fully relaxed until the total energy and maximum ionic Hellmann–Feynman forces converge to $1 \times 10^{-10}$ eV and $1 \times 10^{-4}$ eV/Å, respectively. Lattice parameters of $La_2Zr_2O_7$ (in cubic symmetry, space group $Fd\bar{3}m$) are optimized to be 10.66 Å, in reasonable agreement with experimental data of 10.78 Å[52]. Computational details for $Tl_3VSe_4$ are presented in Supplementary Note 6.

The harmonic and anharmonic interatomic force constants (IFCs) are calculated via the real-space finite displacement difference method, where $2 \times 2 \times 2$ supercells containing 88 atoms are constructed, and Monkhorst–Pack $k$-mesh are set as $3 \times 3 \times 3$. The phonon frequencies and eigenvectors are obtained using the Phonopy[53] package, by diagonalizing the dynamical matrix constructed from the harmonic IFC matrices, and sampling on a $21 \times 21 \times 21$ $q$-mesh. These are typical settings for the lattice dynamics calculations of rare-earth pyrochlore systems[25]. The anharmonic IFCs are obtained using the thirdorder scripts[54]. Interatomic interactions up to the twelfth nearest neighbors (12th NN) are taken into account, corresponding to a cutoff radius ($r^{\text{cutoff}}$) of 7.72 Å; whereas interactions beyond this range are taken to be zero. In fact, increasing the $r^{\text{cutoff}}$ from the 5th NN (corresponding to $r^{\text{cutoff}} = 5.19$ Å) to the 12th NN could sufficiently converge the room-temperature $\kappa_L$ within 7%. It is noteworthy that $La_2Zr_2O_7$ is a complex oxide ceramic with long-range interatomic interactions[55], which might lead to strong dependence of anharmonic IFCs and $\kappa_L$ on the number of neighbor shells. Qin et al.[56] reported that for such material systems, the extent to an adequate inclusion of long-range effect could be estimated by looking into how the interaction strength changes with increased distance between an atomic pair, based on analyzing the root mean square (RMS) of the elements of the harmonic IFC tensor (Frobenius norm):

$$\text{RMS}(\phi_{mn}) = \left[\frac{1}{9} \sum_{\alpha,\beta} \left(\phi_{mn}^{\alpha\beta}\right)^2\right]^{\frac{1}{2}} \tag{12}$$

where $\phi_{mn}$ is the harmonic IFC between atom $m$ and $n$; and $\phi_{mn}^{\alpha\beta}$ is the harmonic response of the force for atom $m$ on the $\alpha$-direction resulted from the displacement of atom $n$ on the $\beta$-direction. Following this approach, the RMS($\phi_{mn}$) for all atomic pairs are analyzed as a function of the interatomic distance. It shows that for $La_2Zr_2O_7$, setting the truncation at the 12th NN is expected to include most strong interatomic interactions up to RMS($\phi_{mn}$) = 2, and beyond this range it decays below RMS($\phi_{mn}$) < 0.15. For these reasons, we chose to include up to the 12th NN interatomic interactions in the present calculations to pursue higher precision.

The lattice thermal conductivity ($\kappa_L^{\text{Phon}}$) is calculated by the iterative solution of the BTE as implemented in the ShengBTE[54] package, with integrations using a $16 \times 16 \times 16$ $q$-mesh. The convergence of $\kappa_L$ with respect to the size of $q$-mesh is tested, and the results show that increasing the $q$-mesh up to $25 \times 25 \times 25$ yields less than 2% difference to the room-temperature $\kappa_L$ of $La_2Zr_2O_7$. Furthermore, non-analytical corrections are applied to the dynamical matrix to take into account long-range electrostatic interactions, based on calculations of Born effective charges ($Z^*$) and dielectric constants ($\varepsilon$) via density functional perturbation theory (DFPT)[57].

The Allen–Feldman calculations are performed with the same $2 \times 2 \times 2$ supercell and $16 \times 16 \times 16$ $q$-mesh, to ensure a consistent comparison between $D_{\text{Phon}}$ vs. $D_{\text{Diff}}^{\text{AF}}$ on a per-mode basis. The Delta function in Eq. (8) is broadened into the Lorentzian form $\frac{\eta/\pi}{(\omega(j)-\omega(p))^2+\eta^2}$. After convergence tests (details in Supplementary Note 7 and Supplementary Fig. 9), the Lorentzian broadening factor $\eta$ is set to be $3.3\Delta_{\text{avg}}$ in our study, where $\Delta_{\text{avg}}$ is the average mode frequency interval ($\Delta_{\text{avg}} = 0.35$ THz for $La_2Zr_2O_7$). Due to the high computational expense, we use per-mode thermal diffusivities on 1000 out of the 4096 wave vectors for our dual-phonon theory under

the Criterion III coupled with Allen–Feldman formula. A convergence test shows that including up to 2000 wave vectors causes only marginal difference (<1%) on the calculated fraction of diffuson-like phonons ($N_{\text{Diff}}/N_{\text{Total}}$) and the final $\kappa_L$ values.

## Data availability
The source data of Figs. 1, 3–5, and Supplementary Figs. 1–9 are provided as a Source data file at https://archive.materialscloud.org/2020.0036/v1, and are further available from the corresponding author upon reasonable request.

## Code availability
Vienna Ab Initio Simulation Package (VASP) is available at www.vasp.at; Phonopy package is available at https://phonopy.github.io/phonopy; ShengBTE code is available at https://bitbucket.org/sousaw/shengbte; thirdorder scripts are available at https://bitbucket.org/sousaw/thirdorder. The custom codes used in this work are available from the corresponding author upon reasonable request.

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

## Acknowledgements

We acknowledge fruitful discussions with Dr. Lucas Lindsay at the Oak Ridge National Laboratory. Y.L. and X.Y. acknowledge the support from the China Scholarship Council. Y.L. acknowledges support from the research project funded by Shenyang National Laboratory for Materials Science. T.F. acknowledges support from the project entitled "Models to Evaluate and Guide the Development of Low Thermal Conductivity Materials for Building Envelopes" funded by Building Technologies Office (BTO), Office of Energy Efficiency & Renewable Energy (EERE) at the Department of Energy (DOE). X.R. acknowledges the partial support from the Defense Advanced Research Projects Agency (Award No. HR0011-15-2-0037).

## Author contributions

X.R. and Y.L. conceived and designed the research. X.R. supervised the research. Y.L. performed the DFT calculations, analyzed the data, prepared the figures and manuscript, with assistance and discussions from X.Y., T.F., and J.W. X.R. and T.F. revised the paper. All authors discussed the results and contributed in writing the paper.

## Competing interests

The authors declare no competing interests.
