## [Peer Review File · Nature Communications]

Reviewers' comments:

Reviewer #1 (Remarks to the Author):

The paper addresses the description of crystalline materials that exhibit a thermal conductivity with glass-like temperature behavior. Namely while in crystal with low anharmonicity and simple (few atoms unit cells) the conductivity decreases at high temperature as $1/T$, in some of the low conductivity crystal (relevant for thermoelectric applications or for thermal isolation) the conductivity decreases with a lower power of T or even increases with temperature as in glassy materials.

The problem is relevant and timely and in the paper is well written and well motivated. The aim of the paper is to arrive to a parameter free prediction of thermal of conductivity in such systems. To this goal the authors introduce a new heuristic approach that they name as “dual-phonon transport theory”.

Such theory has some advantage with respect to other previous heuristic approaches as the ones proposed in ref 11 and 12 of the manuscript. However as in previous approaches it is based on ad hoc assumptions.

At each phonon mode is attributed either a Boltzmann type of heat propagation (BTE type) or a “hopping type” of heat propagation. Three different heuristic criteria are used to decide with type of transport should be applied to a given mode. Interestingly, the authors find that the 3 criteria give similar thermal conductivities.

My major concern is the model of reference [10] (Agne’s formula) used to describe the hopping-type of transport.

In particular in Agne’s formula (eq. 7 of the method section), the diffusivity is proportional to the phonon frequency of the mode ω_j .

The authors of ref. [10] justify this linear heuristic behavior in ω_j quoting incorrectly the results for the diffusivity obtained in glasses with the rigorous (non-heuristic) Kubo-formula approach of Allen and Feldman (PRL 62, 645 (1989)).

Indeed in this paper Allen and Feldman propose that the diffusivity should be of the form [see first column of page 647 of Allen and Feldman PRL 62, 645 (1989)]

$$D(\omega_j) = a^2 \omega_{\text{gabar}} / 3 f(\omega_j / \omega_{\text{gabar}})$$

Where ω_{gabar} is a typical (average) phonon frequency a is the bond characteristic length and $f(x)$ is a adimensional function of the ratio $(\omega_j / \omega_{\text{gabar}})$.

Such a models have been discussed previously by Slack [G. A. Slack, in Solid State Physics, edited by H. Ehrenreich, F. Seitz, and D. Turnbull (Academic, New York, 1979), Vol. 34, p. 1] and Kittel [F. Birch and H. Clark, Am. J. Sci. 238, 529 (1940). 6C. Kittel, Phys. Rev. 75, 972 (1948).]

In Allen and Feldman PRL 62, 645 (1989) fig. 3 suggest to use for $f(x)$ a constant value (see second column at page 647) and fig 3 of the paper. However in a subsequent errata of this paper [Philip B. Allen and Joseph L. Feldman Phys. Rev. Lett. 64, 2466 (1990)] they say that the labels of fig 3 of the PRL 62, 645 (1989) are interchanged. Consequently, the model that best reproduces the Numerical Kubo formula is that in which

$$F(\omega_j / \omega_{\text{gabar}}) = C_2 (\omega_{\text{gabar}} / \omega_j)$$

Where C_2 is an adimensional constant

Namely the diffusivity IS INVERSELY PROPORTIONAL to ω_j and NOT PROPORTIONAL to ω_{gabar} , as INCORRECTLY assumed in Agne's fomula and in the present paper.

Note that even the other model ($f(x) = \text{constant value}$) proposed by Kittel does not have a diffusivity proportional to ω_j as assumed by the authorts of the present manucript (and of ref. [10]).

For these reasons I think that the paper is constructed upon the wrong and unjustified model of Agne. I suggest that the authors reconsider for hopping-type transport the more justified equation validated by Allen and Feldman namely:

$$D(\omega_j) = a^2 \omega_{\text{gabar}}^2 C_2 / (3 \omega_j)$$

Thus I think that the paper cannot be accepted for publication in its present form.

Reviewer #2 (Remarks to the Author):

Thermal transport by atomic vibrations has been described using two different pictures. At the limit of weak perturbation where harmonic normal vibrational modes (phonon) are well defined, the thermal transport can be explained by the Boltzmann transport theory of phonons. At the opposite limit where the perturbation is strong, the thermal transport has been described with the random walk of oscillators. The authors attempt to bridge the gap between the two limits, which is important for understanding low thermal conductivity materials. In this work, they propose to use the Boltzmann transport theory for weakly perturbed modes and the random walk picture for strongly perturbed modes. The authors show that the proposed method gives a better agreement with the experimental thermal conductivity of $\text{La}_2\text{Zr}_2\text{O}_7$ and Ti_3VSe_4 than existing methods.

Although it has some differences from the existing models and gives a good agreement with experimental thermal conductivity of the two materials, the proposed method looks somewhat similar to the two-channel picture proposed by Mukhopadhyay et al. [ref 12] and has similar accuracy. In the lines 65-72, the authors made comments on the issues of the existing models including ref. 12. It is not very clear to me that those issues were satisfactorily resolved in the proposed method as explained below:

1) The authors claim that the scattering rate of well-defined phonons by ill-defined phonons can be calculated using the standard anharmonic lattice dynamics theory. Ref. 15 and 16 are mentioned to support this idea. I could not read ref. 16 because it is not publicly available yet. Ref. 15 shows that the frequencies and linewidths of zone center optical phonons from first principles calculation agree well with the experiment although those phonons have mean free path much smaller than lattice spacing (i.e., ill-defined in space). However, the materials discussed in ref. 15 are very harmonic materials (C, Si, and Ge) and their scattering rate is much smaller than phonon frequency (i.e., well-defined in time). From Fig. 3, many phonon modes in $\text{La}_2\text{Zr}_2\text{O}_7$ have scattering rates already larger than frequency at 300 K (i.e., ill-defined in time). Is the standard anharmonic lattice dynamics theory valid when a vibrational mode involved in the scattering process has larger scattering rate than frequency?

2) The proposed method uses the diffusivity proposed in ref. 10, which I do not think much better than diffusivity models used in ref. 12. I think all those models have similar accuracy and uncertainty because they are based on the simple random walk picture of oscillator or simply assuming lattice constant for a mean free path.

Response to Referees

We have found the reviews to be very useful, and have revised our manuscript accordingly. We believe that the revised version has addressed the reviewer's comments to the best of our ability. A detailed point-by-point response is given below, and revisions are highlighted by yellow bars in the revised manuscript.

Reviewer(s)' Comments to Author:

Reviewer: 1

Comments:

The paper addresses the description of crystalline materials that exhibit a thermal conductivity with glass-like temperature behavior. Namely while in crystal with low anharmonicity and simple (few atoms unit cells) the conductivity decreases at high temperature as $1/T$, in some of the low conductivity crystal (relevant for thermoelectric applications or for thermal isolation) the conductivity decreases with a lower power of T or even increases with temperature as in glassy materials.

The problem is relevant and timely and in the paper is well written and well-motivated. The aim of the paper is to arrive to a parameter free prediction of thermal of conductivity in such systems. To this goal the authors introduce a new heuristic approach that they name as “dual-phonon transport theory”.

Such theory has some advantage with respect to other previous heuristic approaches as the ones proposed in ref 11 and 12 of the manuscript. However as in previous approaches it is based on ad hoc assumptions.

At each phonon mode is attributed either a Boltzmann type of heat propagation (BTE type) or a “hopping type” of heat propagation. Three different heuristic criteria are used to decide with type of transport should be applied to a given mode. Interestingly, the authors find that the 3 criteria give similar thermal conductivities.

My major concern is the model of reference [10] (Agne's formula) used to describe the hopping-type of transport.

In particular in Agne's formula (eq. 7 of the method section), the diffusivity is proportional to the phonon frequency of the mode ω_j .

The authors of ref. [10] justify this linear heuristic behavior in ω_j quoting incorrectly the results for the diffusivity obtained in glasses with the rigorous (non-heuristic) Kubo-formula approach of Allen and Feldman (PRL 62, 645 (1989)).

Indeed, in this paper Allen and Feldman propose that the diffusivity should be of the form [see first column of page 647 of Allen and Feldman PRL 62, 645 (1989)]

$$D(\omega_j) = \frac{1}{3} a^2 \bar{\omega} f(\omega_j/\bar{\omega})$$

Where $\bar{\omega}$ is a typical (average) phonon frequency, a is the bond characteristic length and $f(x)$ is a dimensional function of the ratio $(\omega_j/\bar{\omega})$.

Such models have been discussed previously by Slack [G. A. Slack, in Solid State Physics, edited by H. Ehrenreich, F. Seitz, and D. Turnbull (Academic, New York, 1979), Vol. 34, p. 1] and Kittel [F. Birch and H. Clark, Am. J. Sci. 238, 529 (1940). 6C. Kittel, Phys. Rev. 75, 972 (1948).]

In Allen and Feldman PRL 62, 645 (1989) fig. 3 suggest to use for $f(x)$ a constant value (see second column at page 647) and fig 3 of the paper. However, in a subsequent errata of this paper [Philip B. Allen and Joseph L. Feldman Phys. Rev. Lett. 64, 2466 (1990)] they say that the labels of fig 3 of the PRL 62, 645 (1989) are interchanged. Consequently, the model that best reproduces the Numerical Kubo formula is that in which

$$F(\omega_j/\bar{\omega}) = C_2(\bar{\omega}/\omega_j), \text{ where } C_2 \text{ is a dimensional constant}$$

Namely the diffusivity IS INVERSELY PROPORTIONAL to ω_j and NOT PROPORTIONAL to ω_j , as INCORRECTLY assumed in Agne's formula and in the present paper.

Note that even the other model ($f(x)=$ constant value) proposed by Kittel does not have a diffusivity proportional to ω_j as assumed by the authors of the present manuscript (and of ref. [10]).

For these reasons I think that the paper is constructed upon the wrong and unjustified model of Agne. I suggest that the authors reconsider for hopping-type transport the more justified equation validated by Allen and Feldman namely:

$$D(\omega_j) = \frac{a^2 \bar{\omega}^2 C_2}{3\omega_j}.$$

Thus, I think that the paper cannot be accepted for publication in its present form.

Response: Thank you for the very useful comment. In the revised paper, we have added some results using the Allen-Feldmann formula. The results are in fact consistent with those derived from the random-walk theory, indicating the robustness of our approach. Several points are worthy of clarification.

First, Agne's model is indeed valid in describing the heat conduction of bulk solids in the limit of diffusive transport. It is rooted in the random-walk theory, assuming that heat is transferred through successful random jumps of independent oscillators within a period of oscillation to a distance related with the number density of atoms, and thus results in a "random-walk diffusivity (D_{Diff}^{RW})" proportional to ω . This model has so far been acknowledged as an alternative to the Einstein model, Cahill model, and Allen's model, for understanding heat transfer of solids approaching the amorphous limit. The Allen-Feldmann model, on the other hand, is based on the rigorous Green-Kubo theory, where the "mode diffusivity (D_{Diff}^{AF})" is defined by thermal correlation of each vibrational mode [Allen & Feldmann, Phys. Rev. Lett. 62,6 (1989); Phys. Rev. B, 48, 17 (1993)], which gives a varied $D_{Diff}^{AF} \sim \omega$ dependence.

In our revised manuscript, we calculate the D_{Diff}^{AF} for all vibrational modes of $\text{La}_2\text{Zr}_2\text{O}_7$, by postprocessing our results from harmonic lattice dynamics, and find that D_{Diff}^{AF} initially exhibits a $1/\nu^n$ (parameter n is estimated to be 2 or 4) trend at the low-frequency range; beyond which the frequency dependence becomes not so clear. We added a Supplementary Figure S1 (shown below) to illustrate this point. Such $D_{Diff}^{AF} \sim \omega$ trend is in similar manner with the findings from [Larkin & McGaughey, Phys. Rev. B 89, 144303 (2014); Zhu & Ertekin, Energy Environ. Sci. 12, 216 (2019)], where the ω^{-n} term was interpreted as propagating modes. In addition, κ_L of $\text{La}_2\text{Zr}_2\text{O}_7$ is calculated by using our dual-phonon theory coupled with D_{Diff}^{AF} . The results are consistent with those derived from the random-walk theory, indicating the robustness of our dual-phonon theory. Interestingly, incorporating the random-walk theory arrives at even better agreement with the experimental κ_L of $\text{La}_2\text{Zr}_2\text{O}_7$, in comparison with the

Allen-Feldmann model. Besides, the computational cost for incorporating the analytical formula of Allen-Feldmann model was very high. For $\text{La}_2\text{Zr}_2\text{O}_7$ (22 atoms per primitive unit cell), we needed to work with tremendous amount of data gathered for a $2 \times 2 \times 2$ supercell and 66 branches sampled on $16 \times 16 \times 16$ \mathbf{q} positions. The major expense comes from calculating the cross-correlation velocity that enters into the off-diagonal matrix element of heat current operator. For the sake of computational efficiency and practicality, we only perform limited test with the Allen-Feldmann model.

In the “Results” section of our revised manuscript, we added the new results and discussions on incorporating the Allen-Feldmann model in our dual-phonon theory. Corresponding computational methods are added in the “Methods” section. Figure 1, Figure 3, and Figure 4 are revised accordingly; and Figure S1 is added in the supplementary Information.

Supplementary Figure 1. Thermal diffusivity for each vibration mode calculated from the Allen-Feldmann formula (D_{Diff}^{AF}). Also shown are extrapolations based on an v^{-2} and v^{-4} scaling, corresponding to propagating behaviors.

Reviewer: 2

Comments:

Thermal transport by atomic vibrations has been described using two different pictures. At the limit of weak perturbation where harmonic normal vibrational modes (phonon) are well defined, the thermal transport can be explained by the Boltzmann transport theory of phonons. At the opposite limit where the perturbation is strong, the thermal transport has been described with the random walk of oscillators. The authors attempt to bridge the gap between the two limits, which is important for understanding low thermal conductivity materials. In this work, they propose to use the Boltzmann transport theory for weakly perturbed modes and the random walk picture for strongly perturbed modes. The authors show that the proposed method gives a better agreement with the experimental thermal conductivity of $\text{La}_2\text{Zr}_2\text{O}_7$ and Tl_3VSe_4 than existing methods.

Although it has some differences from the existing models and gives a good agreement with experimental thermal conductivity of the two materials, the proposed method looks somewhat similar to the two-channel picture proposed by Mukhopadhyay et al. [ref 12] and has similar accuracy. In the lines 65-72, the authors made comments on the issues of the existing models including ref. 12. It is not very clear to me that those issues were satisfactorily resolved in the proposed method as explained below:

1. The authors claim that the scattering rate of well-defined phonons by ill-defined phonons can be calculated using the standard anharmonic lattice dynamics theory. Ref. 15 and 16 are mentioned to support this idea. I could not read ref. 16 because it is not publicly available yet. Ref. 15 shows that the frequencies and linewidths of zone center optical phonons from first principles calculation agree well with the experiment although those phonons have mean free path much smaller than lattice spacing (i.e., ill-defined in space). However, the materials discussed in ref. 15 are very harmonic materials (C, Si, and Ge) and their scattering rate is much smaller than phonon frequency (i.e., well-defined in time). From Fig. 3, many phonon modes in $\text{La}_2\text{Zr}_2\text{O}_7$ have scattering rates already larger than frequency at 300 K (i.e., ill-defined in time). Is the standard anharmonic lattice dynamics theory valid when a vibrational mode involved in the scattering process has larger scattering rate than frequency?

Response: Thank you for the good comment. We checked the results and find that the phonons with $1/\tau > \nu$ (ill-defined in time) account for a small fraction of the total vibrational modes. In the figure below, the scattering rates ($1/\tau$) for all vibrational modes of $\text{La}_2\text{Zr}_2\text{O}_7$ and Tl_3VSe_4 are plotted as a function of frequency (ν). The modes that are well-defined in time ($1/\tau < \nu$) could be differentiated from those ill-defined in time ($1/\tau > \nu$). For $\text{La}_2\text{Zr}_2\text{O}_7$, 0.21% of the modes are ill-defined in time at $T=300$ K. They are the low-frequency flat modes contributed by “rattling” motion of La atoms, as illustrated by the phonon dispersion curves in Fig. S2. For Tl_3VSe_4 , 11.58% of modes are ill-defined in time, and they are contributed by “rattling” of the Tl atom. In contrast, we recall that our three criteria yield as high as 75.35%, 23.82% and 65.42% modes respectively as “ill-defined in space” for $\text{La}_2\text{Zr}_2\text{O}_7$ at $T=300$ K; and 74.06%, 44.74% and 66.90% for Tl_3VSe_4 . Therefore, our approach makes a step closer to a sound theory. How to better treat the interactions between the normal phonons and the “ill-defined in time” phonons is a challenging question and can be pursued in the future.

In our revised manuscript, we discussed this point in “Discussion” section, and added a supplementary Figure 9 to support the discussion.

Supplementary Figure 9. Calculated scattering rates of $\text{La}_2\text{Zr}_2\text{O}_7$ and Tl_3VSe_4 from anharmonic lattice dynamics. The modes with $1/\tau < \nu$ and $1/\tau > \nu$ are distinguished in different colors.

2. The proposed method uses the diffusivity proposed in ref. 10, which I do not think much better than diffusivity models used in ref. 12. I think all those models have similar accuracy and uncertainty because they are based on the simple random walk picture of oscillator or simply assuming lattice constant for a mean free path.

Response: In the revised paper, we have added some results using the Allen-Feldmann formula. The results are consistent with those derived from the random-walk theory. To avoid ambiguity, we have removed the arguments on the accuracy and uncertainty.

In summary, the main differences of our approach from existing literature are: (1) We defined the concept of diffuson-like phonons to provide a stronger justification of the phonon scattering calculations of the well-defined phonons (elaborated in detail in our response to the last Comment). (2) Our approach allows for mode-by-mode placement of the phonons, hence avoid the double counting issue in the diffuson-like phonon channel. (3) The three criteria converge to similar results, indicating that the approach is robust. (4) We now show that the diffuson-like phonons could be treated in random-walk picture or the Allen-Feldmann model, which adds to the thoroughness of the theory and robustness of the approach.

Reviewers' comments:

Reviewer #1 (Remarks to the Author):

The authors carefully considered the criticisms that I raised in my previous report. In particular they implemented the Allen-Feldmann formula for diffusivity finding indeed a $1/\omega^n$ with $n>2$. They use the Allen-Feldmann-like diffusivity in their dual phonon model obtaining results not dissimilar from those obtained using the other diffusivity model.

The paper is improved and could be published in Nature Communications after the authors consider and implement the following two suggestions:

- 1) The parameters of the Allen-Feldmann calculation are not described in the method section nor in the additional materials. Some details are reported in the rebuttal letter for reviewers, but not in the paper. In particular the authors should detail the mesh of the phonon momentum (quoting their letter "a $2\times 2\times 2$ supercell and 66 branches sampled on $16\times 16\times 16$ q positions"), the width of the Lorentzian used to replace the delta function, and all other details useful to reproduce fig. S1. Moreover, it would be interesting in showing the dependence of the results of fig. S1 on the somehow "arbitrary" choice of the Lorentzian width.
- 2) From the Allen-Feldmann diffusivity, the authors could compute directly the Allen-Feldmann thermal conductivity without need of using their dual phonon model. The authors should present in the paper or in the additional materials such a result.

Finally, as an optional suggestion, the author could also consider to present, for comparison, in the paper, the conductivity obtained by the theory introduced in [M. Simoncelli, N. Marzari, and F. Mauri, Nature Phys. 15, 809-813 (2019)], Ref. [13] of the manuscript. Indeed, they can obtain the conductivity by a trivial modification of Eqs (8-9) that they use to compute the Allen-Feldmann diffusivity. They should just slightly modify the definition of the off-diagonal velocity (see method section of Nature Phys. 15, 809-813 (2019)) and use, as width of the Lorentzian, the sum of the linewidths (the inverse of the lifetimes presented in Fig 3 of the manuscript) of the two phonons involved in the off-diagonal matrix element.

Reviewer #2 (Remarks to the Author):

The reviewer appreciates the authors for their efforts in the revision. In comparison with recent papers on the new formalism (ref. 13 and 14), the proposed model here still has assumptions and heuristic approaches like the previous models (ref. 10-12). However, the heuristic models have their own value that it can provide some physical insights using conventional (familiar) pictures of phonons and diffusons.

However, the reviewer is still not fully convinced about the use of standard anharmonic lattice dynamics theory for the processes involving phonons that are ill-defined in the time domain. The authors mentioned this is a challenging open question for future study, which the reviewer also agrees with. If the authors could share a bit more details on the scattering processes involving ill-defined modes, the reviewer thinks it would serve as a meaningful dataset. The authors clearly showed in the revision report that the number of modes that are ill-defined in time domain is small; only 0.21% for $\text{La}_2\text{Zr}_2\text{O}_7$ and 11.58 % for Tl_3VSe_4 at 300 K. However, it does not fully disclose the effects of those modes on the scattering rate of well-defined phonons (especially if considering the small scattering rates of well-defined modes.) In the reviewer's opinion, it would be very meaningful if the authors could show how much those modes (ill-defined in time) contribute to the scattering rate of well-defined modes that are described with the Boltzmann transport theory at two temperatures used in Fig. 1 (300 K and 1500 K).

Response to Referees' Comments

We have found the review comments to be very useful, and have revised our manuscript accordingly. We believe that the revised version has addressed the comments to the best of our ability. A detailed point-by-point response is given below, and revisions are highlighted by yellow bars in the revised manuscript.

Reviewer(s)' Comments to Author:

Reviewer #1:

The authors carefully considered the criticisms that I raised in my previous report. In particular they implemented the Allen-Feldmann formula for diffusivity finding indeed a $1/\omega^n$ with $n>2$. They use the Allen-Feldmann-like diffusivity in their dual phonon model obtaining results not dissimilar from those obtained using the other diffusivity model.

The paper is improved and could be published in Nature Communications after the authors consider and implement the following two suggestions:

Response: Many thanks for the overall positive comments.

1) The parameter of the Allen-Feldmann calculation are not described in the method section nor in the additional materials. Some details are reported in the rebuttal letter for reviewers, but not in the paper. In particular the authors should detail the mesh of the phonon momentum (quoting their letter “a $2\times 2\times 2$ supercell and 66 branches sampled on $16\times 16\times 16$ q positions”), the width of the Lorentzian used to replace the delta function, and all other details useful to reproduce fig. S1. Moreover, it would be interesting in showing the dependence of the results of fig. S1 on the somehow “arbitrary” choice of the Lorentzian width.

Response: Thank you for the very useful comment. We added the following calculation details in the “Method” section of our revised manuscript.

“The Allen-Feldmann calculations are performed with the same $2\times 2\times 2$ supercell and $16\times 16\times 16$ q -mesh, to ensure a consistent comparison between D_{Phon} vs. D_{Diff}^{AF} on a per-mode basis. The Delta function in Eq. (8) is broadened into the Lorentzian form $\frac{\eta/\pi}{(\omega(j)-\omega(p))^2+\eta^2}$. After convergence tests (details in

Supplementary Note 7 and Supplementary Figure 9), the Lorentzian broadening factor η is set to be $3.3\Delta_{avg}$ in our study, where Δ_{avg} is the average mode frequency interval ($\Delta_{avg} = 0.35$ THz for $\text{La}_2\text{Zr}_2\text{O}_7$). Due to the high computational expense, we use per-mode thermal diffusivities on 1000 out of the 4096 wave vectors for our dual-phonon theory under Criterion III coupled with Allen-Feldmann formula. A convergence test shows that including up to 2000 wave vectors causes only marginal difference ($<1\%$) on the calculated fraction of diffuson-like phonons (N_{Diff}/N_{Total}) and the final κ_L values.”

We also added Supplementary Note 7 with the following content:

“We did a convergence test on the Lorentzian broadening factor using a range from $1.1\Delta_{avg}$ to $4.4\Delta_{avg}$ for some wave vectors. The Supplementary Figure 9 shows the calculated frequency-dependent D_{Diff}^{AF} of $\text{La}_2\text{Zr}_2\text{O}_7$ with Lorentzian broadening factors of $2.2\Delta_{avg}$, $3.3\Delta_{avg}$, and $4.4\Delta_{avg}$, for the modes at $\mathbf{q}=(0.25, 0.25, 0.25)$ and $\mathbf{q}=(0.5, 0.5, 0.5)$. We observe reasonable convergence. Moreover, the uncertainties of the calculated κ_L^{AF} are less than 6.2% for $\mathbf{q}=(0.25, 0.25, 0.25)$ and 5.6% for $\mathbf{q}=(0.5, 0.5, 0.5)$, respectively. Note that the data for $1.1\Delta_{avg}$ have not converged yet, and thus is not shown in the figure. Hence, we use a

broadening factor of $3.3\Delta_{\text{avg}}$ in our calculations. We also note that it has been acknowledged that, a frequency-dependent Lorentzian broadening width may be useful [M. Simoncelli, et al. Nature Phys. 15, 809-813 (2019); J. Larkin, et al. Phys. Rev. B 89, 144303 (2014)].”

We added the Supplementary Figure 9 as follow:

Supplementary Figure 9. The calculated frequency-dependent D_{Diff}^{AF} of $\text{La}_2\text{Zr}_2\text{O}_7$ with Lorentzian broadening factors of $2.2\Delta_{\text{avg}}$, $3.3\Delta_{\text{avg}}$ and $4.4\Delta_{\text{avg}}$, for the modes at $q=(0.25, 0.25, 0.25)$ and $q=(0.5, 0.5, 0.5)$.

2) From the Allen-Feldmann diffusivity, the authors could compute directly the Allen-Feldmann thermal conductivity without need of using their dual phonon model. The authors should present in the paper or in the additional materials such a result.

Response: Thank you for the good comment. In our revised manuscript, κ_L values calculated directly from the Allen-Feldmann formulism are added.

We added a Supplementary Note 2 with the following content:

“ κ_L could be directly calculated from $\kappa_L^{AF} = \sum_{j=1}^{3N} C_s(j) D_{Diff}^{AF}(j)$, where $C_s(j)$ is the per-mode specific heat (see Eq. (4) in the main text), and j sums over all the vibrational modes (N denotes the number of atoms in the simulation unit cell). Results of calculated κ_L^{AF} are shown in the inset of Supplementary Figure 1. This could be seen as the “amorphous limit” for solids. Herein, the ab-initio derived thermal diffusivity is temperature independent, and thus the trend of κ_L^{AF} is dominated by temperature dependence of specific heat.”

Also, the Supplementary Figure 1 (see below) is revised to include the κ_L data in the inset.

Supplementary Figure 1. The thermal diffusivity (D_{Diff}) for each vibrational mode of $\text{La}_2\text{Zr}_2\text{O}_7$ calculated from the Allen-Feldmann formula. Also shown are extrapolations based on an ν^{-2} and ν^{-4} scaling. Inset: Temperature-dependent thermal conductivity (κ_L) calculated directly from the Allen-Feldmann theory.

Finally, as an optional suggestion, the author could also consider to present, for comparison, in the paper, the conductivity obtained by the theory introduced in [M. Simoncelli, N. Marzari, and F. Mauri, *Nature Phys.* 15, 809-813 (2019)], Ref. [13] of the manuscript. Indeed, they can obtain the conductivity by a trivial modification of Eqs (8-9) that they use to compute the Allen-Feldmann diffusivity. They should just slightly modify the definition of the off-diagonal velocity (see method section of *Nature Phys.* 15, 809-813 (2019)) and use, as width of the Lorentzian, the sum of the linewidths (the inverse of the lifetimes presented in Fig 3 of the manuscript) of the two phonons involved in the off-diagonal matrix element.

Response: Thank you for the very useful advice and guidance on applying the theory of [M. Simoncelli, N. Marzari, and F. Mauri, *Nature Phys.* 15, 809-813 (2019)]. Indeed, we are very interested in comparing our results with those with their approach, together with considering how to combine the mathematical rigor of their approach and the physical insights from the heuristic models to further the understanding of this problem. However, considering the calculations are very expensive and our paper is already quite long, we will definitely pursue this in the next effort. We have added the following sentence at the end of the “Discussion” section:

“Future studies can include a comparison of our approach with those in Refs. [13-14], a consideration of how to combine the mathematical rigor of those approaches and the physical insights from our approach and other heuristic models, ...”

Reviewer: 2

Comments:

The reviewer appreciates the authors for their efforts in the revision. In comparison with recent papers on the new formalism (ref. 13 and 14), the proposed model here still has assumptions and heuristic approaches like the previous models (ref. 10-12). However, the heuristic models have their own value that it can provide some physical insights using conventional (familiar) pictures of phonons and diffusons.

However, the reviewer is still not fully convinced about the use of standard anharmonic lattice dynamics theory for the processes involving phonons that are ill-defined in the time domain. The authors mentioned

this is a challenging open question for future study, which the reviewer also agrees with. If the authors could share a bit more details on the scattering processes involving ill-defined modes, the reviewer thinks it would serve as a meaningful dataset. The authors clearly showed in the revision report that the number of modes that are ill-defined in time domain is small; only 0.21% for $\text{La}_2\text{Zr}_2\text{O}_7$ and 11.58 % for Tl_3VSe_4 at 300 K. However, it does not fully disclose the effects of those modes on the scattering rate of well-defined phonons (especially if considering the small scattering rates of well-defined modes.) In the reviewer's opinion, it would be very meaningful if the authors could show how much those modes (ill-defined in time) contribute to the scattering rate of well-defined modes that are described with the Boltzmann transport theory at two temperatures used in Fig. 1 (300 K and 1500 K).

Response: Thank you for the insightful comment. Indeed, this will be very useful information. We have now calculated these results and added into the paper. In the "Discussion" section we have added:

"As shown in Fig. 5(a) and (c), the number of modes that are ill-defined in the time scale ($1/\tau > \nu$) account for 0.21% for $\text{La}_2\text{Zr}_2\text{O}_7$ and 11.58% for Tl_3VSe_4 at $T=300$ K, which are much smaller than the percentages of the ill-defined-in-space phonons. They rise to 45.36% at $T=1500$ K for $\text{La}_2\text{Zr}_2\text{O}_7$ and 59.71% at $T=500$ K for Tl_3VSe_4 , respectively, though. We then further evaluate the effects of these modes on the scattering rates of phonons that are well-defined in the time scale ($1/\tau < \nu$), by calculating the percentages of scattering rates of well-defined-in-time modes contributed by three-phonon scattering processes involving at least one ill-defined-in-time mode. Fig. 5(b) and (d) show that the percentages are mostly below 20% for $\text{La}_2\text{Zr}_2\text{O}_7$ and below 60% for Tl_3VSe_4 at $T=300$ K; which rise to mostly above 70% at $T=1500$ K for $\text{La}_2\text{Zr}_2\text{O}_7$, and above 90% at $T=500$ K for Tl_3VSe_4 . Therefore, although our approach makes a step closer to a sound theory, and appears to agree with experiments, whether we can treat scattering by these ill-defined-in-time phonons with standard anharmonic lattice dynamics still remains a challenging open question, especially at high temperatures. Future studies are warranted on how to better understand the interactions between the "well-defined-in-time" phonons and "ill-defined-in-time" phonons in our model and other lately developed models [M. Simoncelli, et al. Nature Phys. 15, 809-813 (2019); L. Isaeva, et al. Nat. Commun. 10: 3853 (2019)]."

We also added the Fig. 5 in our revised manuscript.

Fig. 5. (a) and (c) The scattering rates ($1/\tau$) of $\text{La}_2\text{Zr}_2\text{O}_7$ and Tl_3VSe_4 calculated from standard anharmonic lattice dynamics. The results at $T=300$ K and $T=1500$ K are presented for $\text{La}_2\text{Zr}_2\text{O}_7$; and $T=300$ K and $T=500$ K for Tl_3VSe_4 . The vibrational modes having $1/\tau < \nu$ (well defined in the “time scale”) and $1/\tau > \nu$ (ill-defined in the “time scale”) are distinguished in different colors; and the baseline $1/\tau = \nu$ is plotted as a guide for the eye. (b) and (d) The percentages of scattering rates of well-defined-in-time modes contributed by three-phonon scattering processes involving at least one ill-defined-in-time mode.

REVIEWERS' COMMENTS:

Reviewer #1 (Remarks to the Author):

The authors implemented all the suggestions of the referees. For me the paper can be accepted for publication.

Reviewer #2 (Remarks to the Author):

The authors addressed all of my concerns. I recommend its publication.

Response to Referees' Comments

We thank the reviewers for the positive comments and the recommendations for publication.

Reviewer(s)' Comments to Author:

Reviewer #1:

Comments: The authors implemented all the suggestions of the referees. For me the paper can be accepted for publication.

Response: Many thanks for the positive comments!

Reviewer #2:

Comments: The authors addressed all of my concerns. I recommend its publication.

Response: Many thanks for the positive comments!